# Optimized Fertilization Shifted Soil Microbial Properties and Improved Vegetable Growth in Facility Soils with Obstacles

Yiqian Lang [1,2], Yun Ma [1], Guiliang Wang [1], Xiaoqing Qian [1,2] and Juanjuan Wang [1,2,*]

1    Department of Resources and Environmental Science, College of Environmental Science and Engineering, Yangzhou University, Yangzhou 225127, China; 221606105@stu.yzu.edu.cn (Y.L.); malu1210@163.com (Y.M.); wgl0520@yzu.edu.cn (G.W.); qianxq@yzu.edu.cn (X.Q.)
2    Key Laboratory of Arable Land Quality Monitoring and Evaluation, Yangzhou University, Ministry of Agriculture and Rural Affairs, Yangzhou 225127, China
*    Correspondence: wangjuanjuan@yzu.edu.cn

**Abstract:** Currently, facility cultivation produces almost a third of all vegetables in China. The intensive production style has led to serious soil problems that need to be tackled. In this paper, a pot experiment was set up to evaluate the effects of optimized fertilization on vegetable growth and soil properties. Specifically, calcium, magnesium, boron and molybdenum were added on the basis of soil testing. The results showed that the growth of Chinese cabbage was significantly increased by optimized fertilization. The soil pH increased (by 3.82%), and EC decreased (by 8.54%). The abundance of culturable bacteria increased by 33.86%, whereas that of fungi decreased by 70.7%. The optimized fertilization increased the richness but not the evenness of soil microorganisms, increased the relative abundance of Proteobacteria, Firmicutes, Bacillus and Bacteroidetes, and decreased the relative abundance of Actinobacteria, Chloroflexi, and superphyla. Optimized fertilization inhibited the growth of Chytridiomycota and Mortierellomycota, especially the plant pathogen *Fusarium*. Moreover, balanced fertilization was beneficial in promoting various metabolic processes of soil bacteria. Soil water-soluble Ca, Mg, and available Mo might be the main factors driving the change in microbial groups.

**Keywords:** facility soil; soil obstacles; balanced fertilization; Chinese cabbage; microbial community

## 1. Introduction

China is the largest vegetable producer globally [1], and the vegetable planting area in plastic sheds has reached 2.85 million hectares by 2023, accounting for 81% of total vegetable production (National Bureau of Statistics, 2023). Jiangsu Province is located in the middle and lower reaches of the Yangtze River, with an optimal climate, and the facility vegetable production area of Jiangsu province accounts for more than 20%, ranking the first in the country [2].

On a global scale, soil fertility and soil ecosystem sustainability are severely affected in facility production [3,4]. With a high multiple cropping index, the greenhouse cultivation adopts a highly intensive mode, and there are problems such as excessive fertilization and exploitation of soil, accompanied with salt and nutrient accumulation [5,6]. These will lead to a reduction in yield and quality of vegetables eventually. Moreover, excess nutrients in the soil may pollute the environment, including surface and groundwater, and lead to significant greenhouse gas production [7,8]. The fertilizer applied in the facility soil is much higher than that in the open soil, especially the application of nitrogen (N) fertilizer. It is estimated that the residual rate of nitrogen fertilizer in the installed soil reached 53% in the Yangtze River Delta region of China [9]. As compared with open soil, the soil pH value decreased by 3.9%, whereas the soil effective NPK increased by 62.7%, respectively [10]. Long-term greenhouse cultivation increased soil acidity, salinity, as well as nitrate accumulation in Jiangsu province [11,12].

In Europe, two-season greenhouse planting uses an average of 650 kg N per hectare [13], whereas it is more than 2000 kg N per hectare in China, 4–7 times the crop needs [14]. The lack of accurate fertilization caused the imbalance of soil nutrients in vegetable greenhouses. The content of soluble salt in the soil of vegetable field in Yangtze River Basin was 3.08 g kg$^{-1}$, much higher than that of open-air cultivation [10]. According to our previous investigation, there were imbalances of calcium, magnesium, boron, manganese, iron, zinc, molybdenum, and other elements in the facility soil in Jiangsu province. Therefore, it is critical to apply formula fertilization on the basis of soil analyses, given there must be a balance between medium and trace element fertilizers and NPK. It is common to have an accumulation of certain nutrients such as nitrogen, phosphorus (P), and heavy metal elements [11] but a scarcity of other nutrients [15,16] in facility soils.

Soil microbial community composition is particularly important for maintaining a healthy ecosystem [17]. Soil microbes, important indicators of soil health, participate in the formation of soil structure, decomposition of organic matter, nutrient cycling, and the improvement of plant resistance [18,19]. The common problems, including secondary salinization, acidification, and high nutrient levels in facility soil, have a great impact on soil microbial activity, diversity, and community composition [20]. The imbalance of soil nutrients will directly affect the soil microbial ecosystem, resulting in an imbalance between beneficial and harmful microorganisms [21–23]. Soil acidification in facilities could decrease bacterial diversity while increasing fungal diversity [24]. Soil electrical conductivity (EC) is another dominant factor affecting the composition of bacterial communities in secondary salinized soils [25]. Studies have shown that the activity and diversity of soil microorganisms decreased with the increase in soil EC [26]. The composition of microbial community was also affected by soil secondary salinization. With the increase in soil EC, the relative abundance of Bacteroides decreased, whereas Acidobacteria increased [4]. The negative conditions resulted in a significant decrease in the abundance of beneficial bacteria such as Nitrospirillum and Bacillus [27]. On the other hand, continuous cropping in facility agriculture may decrease fungal diversity with an increase in pathogenic fungi [28,29], adding to soil deterioration.

Among others, the most effective method is to balance fertilization. According to the 4R Plant Nutrition Manual [30], the optimal fertilization strategy is to provide the right nutrients to the plant roots at the right time, in the right place, with the right proportions. Optimized fertilization can improve soil fertility [31], rebuild soil ecosystem, increase soil microbial biomass, activity and diversity, regulate microbial community structure [32], improve soil enzyme activity and bacterial quantity, and ultimately increase crop yield [33]. In addition, as most farmers lack the knowledge of precise fertilization, a large amount of fertilizer is input in facility agriculture, and excessive fertilization causes a series of soil problems [34]. For example, in Shouguang City, Shandong Province, one of China's major greenhouse vegetable producing areas, the soil pH dropped from 8.39 to 7.38, whereas the electrical conductivity increased nearly twice [35]. The content of soil soluble salt in greenhouse was nearly four times that of open-air cultivation in Shanghai. The ratio of N, P, and potassium (K) in facility soil is severely out of balance.

Therefore, it becomes the primary task to achieve balanced soil fertilization conditions, meaning overall improved soil ecosystems in facility production. This experiment collected soils with serious obstacles in a vegetable base near Yangzhou City, Jiangsu Province, and designed a balanced fertilization scheme based on soil properties, aiming to investigate the influences of optimized fertilization on the improvement of soil conditions.

## 2. Materials and Methods

### 2.1. Experimental Materials

The soil used in the experiment was taken from the vegetable base of Yizheng City, Yangzhou, Jiangsu Province. The soil was sandy loam, belonging to the Calcareous alluvial soil developed from the alluvial parent material of the Yangtze River. Due to long-term continuous facility cultivation, the soil was deteriorating, which negatively affected the

vegetable yield and quality. The basic properties of soil were determined before the pot experiment: pH 5.52, exchangeable acid 0.578 cmol kg$^{-1}$, EC 0.884 ms cm$^{-1}$, organic matter 40.9 mg kg$^{-1}$, ammonium nitrogen 39.7 mg kg$^{-1}$, nitrate nitrogen 105.1 mg kg$^{-1}$, available phosphorus 174.3 mg kg$^{-1}$, and rapidly available potassium 310 mg kg$^{-1}$. Water-soluble Ca and Mg were 25.7 and 11.2 mg kg$^{-1}$, respectively.

The vegetable variety tested was *Brassica chinensis* L.

### 2.2. Experimental Setup

According to the pre-determination of soils, an optimized fertilization scheme was set up, with a targeting soil pH of 6.5, exchangeable calcium and magnesium 1200 and 300 mg kg$^{-1}$, respectively, and available molybdenum and boron were 0.3 and 3.0 mg kg$^{-1}$, respectively.

A pilot experiment was conducted to test the effects of fertilization on the growth of Chinese cabbage. The treatment with the best growth performance was chosen as the optimized treatment (OFT) for further analyses of soil characteristics. This treatment included: calcium oxide 0.429 g kg$^{-1}$ soil, magnesium oxide 0.139 g kg$^{-1}$ soil, ammonium molybdate 0.0001 g kg$^{-1}$ soil, and borax 0.011 g kg$^{-1}$ soil.

PVC boxes (305 × 200 × 90 mm) were used for the cultivation in the experiment. For the OFT treatment, the above-mentioned fertilizers were prepared in stock solution, then diluted and mixed with 5 kg of dry soil using the Fourfold Method, to make sure all fertilizers were mixed thoroughly. A control (CK) without any fertilizer was prepared. Then, the soils were compacted in the box and watered with distilled water, balanced for three days before sowing. In total, 100 Chinese cabbage seeds were evenly sown in each box and cultured in a greenhouse at 28~30 °C. Both the OFT treatment and the control were made in triplicates.

During the growth, the soil moisture content was maintained. The growth period was from 25 September 2022 to 28 October 2022.

### 2.3. Sample Analyses

After the harvest, the soil on the root surface of cabbages were cleaned with deionized water, blotted dry with absorbent paper, and weighed. The samples were then placed in a brown paper bag and dried in an oven at 105 °C for 15 min to kill the green, then dried at 70 °C to a constant weight (about 48 h). The dry biomass of the plants was weighed, and the moisture content was calculated.

The soil from each box was mixed, and three portions were taken for the determination of soil culturable microorganisms, microbial community composition, and basic chemical properties, respectively.

#### 2.3.1. Determination of Soil Chemical Properties

The major soil chemical properties were measured following the standard testing procedures [36]. Among them, the soil moisture content was determined using drying weight loss method; soil organic matter (OM) using potassium dichromate volumetric method and external heating method; soil alkali-hydrolyzed nitrogen using alkali-hydrolytic method; soil nitrate nitrogen and ammonium nitrogen were extracted by 2 mol L$^{-1}$ KCl and determined photometrically; soil available phosphorus using Mo-Sb colorimetric method; soil available potassium by flame spectrophotometry; and water-soluble calcium (Ca) and magnesium (Mg) were extracted by deionized water, titrated by EDTA. The available boron (B) and molybdenum (Mo) in soil were determined by DTPA extraction and inductively coupled plasma mass spectrometry (ICP-MS).

#### 2.3.2. Determination of Soil Culturable Microorganisms

Beef extract peptone solid medium and PDA medium were used to count the culturable bacteria and funga, respectively. The brief steps were as follows: 0.5 g of fresh soil samples from different treatments was diluted in 50 mL sterile water and shaken

at 150 r min$^{-1}$ for 30 min, before series dilution. An aliquot of 100 μL dilution of three concentrations ($10^{-4}$, $10^{-5}$, and $10^{-6}$) were used as inoculum for spreading plat on beef extract peptone (bacteria) or PDA (fungi). A minimum of 3 repetitions were prepared for each dilution. The plates were incubated in the dark at 25 °C for two to three days. The colonies growing on the plates of different dilutions were counted, and colony forming units (CFUs) of culturable bacteria or fungi in the soil was calculated.

2.3.3. Analysis of Soil Microbial Community Structure

Soil genomic DNA was extracted using MoBio PowerSoil DNA Isolation DNA extraction kit (QIAGEN Inc., Valencia, CA, USA) according to the instructions. The quality of soil DNA extracted was verified by gel electrophoresis in 1% agarose gel. The V3-V4 fragments of the bacterial 16S rRNA gene were amplified by PCR using primers 338F (5′-ACTCCTACGGGAGCAGCAG-3′) and 806R (5′-GGACTACHVGGGTWTCTAAT-3′). The PCR reaction consisted of 25 μL of the following mixture: 10 μL of ultrapure water, 5 μL of 5 × FastPfu buffer, 2 μL of 2.5 mM dNTPs, 1 μL each of the two primers, 0.5 μL of FastPfu polymerase, 10 ng of soil DNA, and 0.25 μL of bovine serum albumin. Amplification conditions were as follows: DNA denaturation at 94 °C for 3 min, followed by 30 amplification cycles (94 °C 45 s, 50 °C 45 s, and 72 °C 45 s), and finally extended at 72 °C for 10 min. Soil fungal community DNA were amplified with primers ITS1F(5′-CTTGGTCATTTAGAGGAAGTAA-3′) and ITS1R(5′-GCTGCGTTCTTCATCGATGC-3′), with the amplification conditions: Denaturation at 98 °C for 1 min, followed by 30 amplification cycles (denaturation at 98 °C for 10 s, annealing at 50 °C for 30 s, extension at 72 °C for 60 s), and then a final extension at 72 °C for 5 min. Amplification results were examined with 2% agarose gel, and the amplified PCR products were purified and quantified using AxyPrepDNA gel (Promega, Madison, WI, USA). Finally, high-throughput sequencing was performed on the Illumina MiSeq platform (Shanghai Majorbio Bio-pharm Technology Co., Ltd., Shanghai, China).

Bioinformatic analysis of soil bacterial or fungal sequences was performed on the Majorbio I-Sanger platform (http://www.i-sanger.com, accessed on 23 October 2023). First, the sequences obtained by Illumina Miseq were coarsely analyzed using Mothur software package (Mothur, 1.30.2). Sequences that did not match barcode or less than 250 base pairs were filtered; the chimera was detected and removed by Uchime. The selected sequences were then compared with the Silva ribosomal RNA database for bacteria and the UNITE ITS database for fungi. Assignment of operational classification units (OTUs) was based on 97% sequence similarity level. Alpha ($\alpha$) diversities of bacteria and fungi, including Sobs, ACE, Chao, Shannon, and Simpson indices, were calculated using the corresponding formulas on Major I-Sanger platform. Venn diagram analysis was used to count the number of common and unique species among the treatments. According to the results of database comparison, the ratio of bacteria and fungi in different classification levels and the proportion of dominant species in the soil samples were calculated.

LEfSe analysis was used to identify the groups with significant differences in abundance, and linear discriminant analysis (LDA) was used to estimate the influence of each component (taxa) on the difference. The Spearman correlation method was used to analyze the correlation between bacteria or fungi and soil physicochemical properties. Finally, the predictive PICRUSt software (PICRUSt2, 2.2.0) was used to make functional prediction of the soil bacterial sequencing results, i.e., the OTU abundance table was standardized through PICRUSt, and then the OTU was annotated by KEGG function according to the corresponding greengene id of each OTU. The annotation information of OTU at each function level in KEGG and the abundance information of each function in different samples were obtained, and the abundance of related metabolic pathways was compared.

*2.4. Statistical Analysis*

The results of soil physical and chemical properties were analyzed and plotted by Excel 2016. The differences of soil properties, bacterial and fungal CFU, as well as vegetable

biomass in OFT treatment and the control were compared using Independent samples T test in SPSS 26.0 software (IBM Corp., Armonk, NY, USA).

## 3. Results

### 3.1. Effects of Different Treatments on Soil Basic Chemical Properties

The effects of optimal fertilization treatments on soil pH, conductivity, and nutrients are listed in Table 1. As can be seen from the table, the soil pH increases, and EC decreased slightly after the treatment. There was little difference observed in soil organic matter and soil nitrate, whereas soil available B and available Mo increased ($p < 0.05$) after the treatment of optimized fertilization. There was a certain increase in soil water-soluble Ca and Mg contents, but this was not significant due to a large variation.

**Table 1.** Major soil physicochemical properties soil with/without optimized fertilization schemes.

| Treatment | pH | EC ($\mu$S cm$^{-1}$) | OM (g kg$^{-1}$) | NN (mg kg$^{-1}$) | Ca (mg kg$^{-1}$) | Mg (mg kg$^{-1}$) | B (mg kg$^{-1}$) | Mo (mg kg$^{-1}$) |
|---|---|---|---|---|---|---|---|---|
| CK | 5.76 ± 0.14 | 894.8 ± 25.8 | 37.9 ± 1.18 | 41.9 ± 6.89 | 10.9 ± 3.0 | 6.4 ± 1.89 | 1. 0 ± 0.19 | 0.23 ± 0.01 |
| OFT | 5.98 ± 0.30 | 764.7 ± 18.46 | 38. 3 ± 1.29 | 40.71 ± 5.01 | 15.3 ± 1.94 | 7.2 ± 0.97 | 1.50 ± 0.15 | 0.25 ± 0.02 |

EC: electrical conductivity; NN: nitrate-N; Ca: water-soluble calcium; Mg: water-soluble magnesium; B: available boron; Mo: available molybdenum.

### 3.2. Effects of Formula Fertilization Treatment on Culturable Microorganisms in Soil

The results of culturable microorganisms obtained by the dilution plate method are shown in Figure 1. The culturable bacteria in the soil increased from $4.43 \times 10^5$ to $5.93 \times 10^5$ CFU g$^{-1}$ soil after the optimized fertilization treatment, with an increase of 33.86%. The culturable fungi in soil treated formula fertilization was significantly lower ($9.98 \times 10^4$ CFU g$^{-1}$ soil) in the treatment than in the control ($3.41 \times 10^5$ CFU g$^{-1}$ soil) ($p < 0.05$).

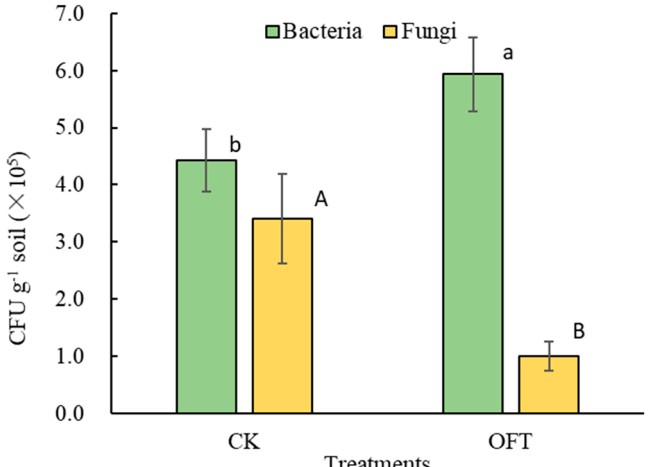

**Figure 1.** The colony formation unit of soil culturable bacteria and fungi in CK and OFT treatment. Different initial/capital letters indicate significant differences among the treatments in bacterial or fungal CFU, individually.

### 3.3. Effects of Optimized Fertilization on Soil Microbial Community Shift

According to the soil physicochemical indexes and the results of soil culturable microorganisms, six samples of the treatment and control were analyzed for bacterial and fungal community structure. In total, 11,547 bacterial OTUs and 3436 fungal OTUs were obtained. For fungi, six samples were treated in two treatments, with a total of 218,930 sequences, with an average length of 246 bp.

### 3.3.1. Soil Microbial Diversity

The calculated results of soil microbial diversity for each treatment are listed in Table 2. The bacterial community richness was estimated by ACE, Sobs, and the Chao index, while bacterial community evenness was expressed by the Shannon index and Simpson index. The results showed that the richness of bacteria and fungi in soil increased after optimized fertilization treatment. The diversity index of bacteria changed little, the Simpson index of fungi increased by 17.1% after treatment, and the Shannon index remained unchanged. The coverage rate of all samples was higher than 98%, indicating that the sequencing results were representative.

**Table 2.** Bacterial and fungal Alpha diversity indexes of balanced fertilization treatment and control soil.

| | Treatment | Sobs Index | ACE Index | Chao Index | Shannon Index | Simpson Index | Coverage% |
|---|---|---|---|---|---|---|---|
| Bacteria | CK | 1038 ± 113.2 | 1356.6 ± 150.8 | 1356.2 ± 145.8 | 5.42 ± 0.16 | 0.012 ± 0.003 | 98.42 |
| | OFT | 1107.7 ± 89.76 | 1492.8 ± 103.2 | 1494.5 ± 141.4 | 5.34 ± 0.12 | 0.015 ± 0.001 | 98.17 |
| Fungi | CK | 167.7 ± 22.2 | 191.3 ± 29.7 | 191.3 ± 25.5 | 2.9 ± 0.3 | 0.105 ± 0.03 | 99.90 |
| | OFT | 181.7 ± 12.9 | 210.9 ± 12.5 | 208.4 ± 9.41 | 2.9 ± 0.21 | 0.118 ± 0.038 | 99.88 |

### 3.3.2. Soil Microbial Community Composition

A Venn diagram based on OTU numbers was created to visualize the difference in microbial composition between optimized fertilization treatment and the control (Figure 2). There were 518 (24.92%) and 392 (19.86%) unique bacterial OTUs in treatment and control, respectively, whereas there were 1169 OTUs shared by the two, including the most abundant OTU. After balanced fertilization treatment, the soil bacterial diversity increased significantly, with 126 more OTUs than control. For soil fungi, there were 346 OTUs, of which 74 and 70 unique OTUs were for treatment and control, respectively. The number of soil fungal OTU was not affected by the optimized fertilization treatment.

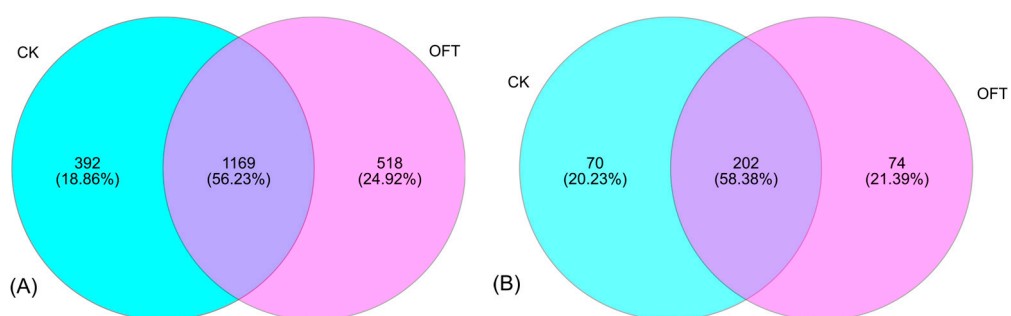

**Figure 2.** Venn diagram based on OTU of the treatment and control ((**A**). Bacteria; (**B**). Fungi). Blue color represents control, and pink color represents optimized fertilization treatment (OFT).

Based on 16S rRNA sequences, a total of 58 bacterial phyla were retrieved. The dominant bacteria phyla (Figure 3A) in the treatments included Proteobacteria, Actinobacteriota, Chloroflexi, Firmicutes, Patescibacteria, Gemmatimonadota, and Bacteroidota, accounting for 94.66% and 92.22% of total bacterial in balanced fertilization treatment and control, respectively. Among the dominant phyla, Proteobacteria, Firmicutes, Gemmatimonadota, and Bacteroidetes increased 14.37%, 23.64%, 105.2%, and 26.22%, respectively, compared with the control. Actinobacteriota, chloroflexi, and superphyla decreased by 8.13%, 38.78%, and 40.42%, respectively.

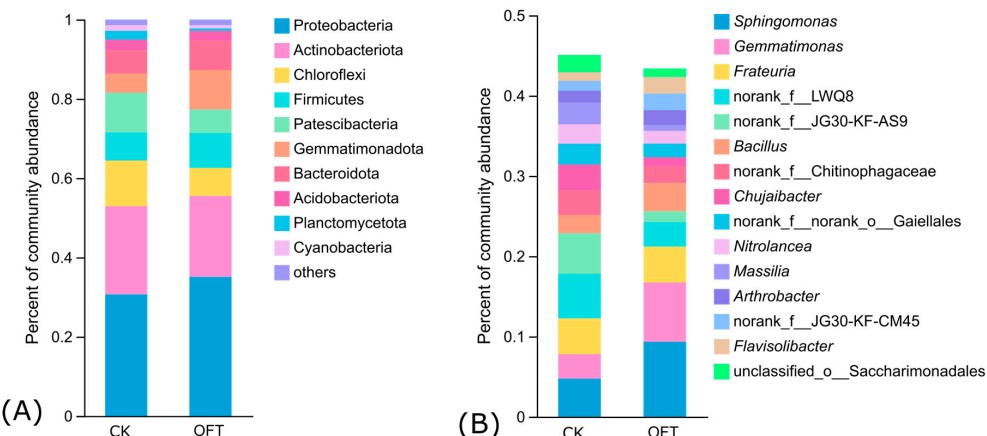

**Figure 3.** Relative abundance of major bacterial groups at phylum (**A**) and genus level (**B**) in soil treated with balanced fertilization and control (only those with relative abundancy >1% are listed).

At the genus level (Figure 3B), the genus *Sphingomonas* belonging to Proteobacteria had the highest abundance, reaching 9.33% in optimized fertilization treatment, 4.74% higher than that in control. Similarly, *Gemmatimonas*, the second most abundant genus occupied 7.39%, 3.05% higher than that of the control. The third abundant *Frateuria* was not significantly different between the treatment and control, whereas the unknown genera norank_f__LWQ8 and norank_f__JG30-KF-AS9, belonging to Superphyla and chloroflexi, respectively, were more abundant in the control soil.

The community composition based on the results of fungal ITS sequence determination is shown in Figure 4. As can be seen from the figure, Ascomycota was the most dominant fungi in each soil, accounting for 79.38% and 73.33% in the optimized fertilization treatment and the control soil, respectively. The second most abundant taxa were Basidiomycota (16.57% and 14.81% in treatment and control, respectively). The abundances of Chytridiomycota and Mortierellomycota were higher in control samples and were 228.57% and 277.78% of those of the optimized treatment, respectively, indicating that balanced fertilization treatment inhibited the growth of these phyla.

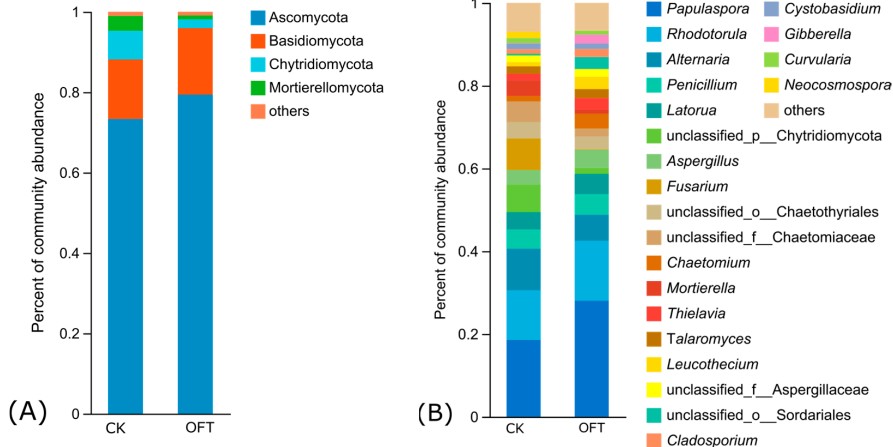

**Figure 4.** Dominant fungal groups at phylum (**A**) and genus level (**B**) in balanced fertilization treatment and control soils.

At the genus level, *Papulaspora* represented the most abundant fungal genus, followed by *Rhodotorula*. Both of these genera were more abundant in OFT treatment than in control. *Fusarium*, a typical plant pathogenic fungus, was found to be significantly abundant in CK (7.63%) as compared with OFT treatment (0.11%).

The results of LEfSe multilevel discriminant analysis are shown in Figure 5. The same color nodes represent the microbial groups that are significantly enriched in the corresponding group and have a significant impact on the treatment difference. The light-yellow nodes represent microbial groups that do not differ significantly between treatments. The diameter of each circle is proportional to the abundance of each taxon. As can be seen from Figure 5, at the phylum level, Bacteroidota, Gemmatimonadota, Bdellovibrionota, and Sumerlaeota were significantly enriched in optimized fertilization treatment and contributed significantly to the differences between treatments and control. The phylum Chloroflexi and WPS-2 were biomarkers for the control. After optimized fertilization treatment, 52 bacterial groups were significantly enriched.

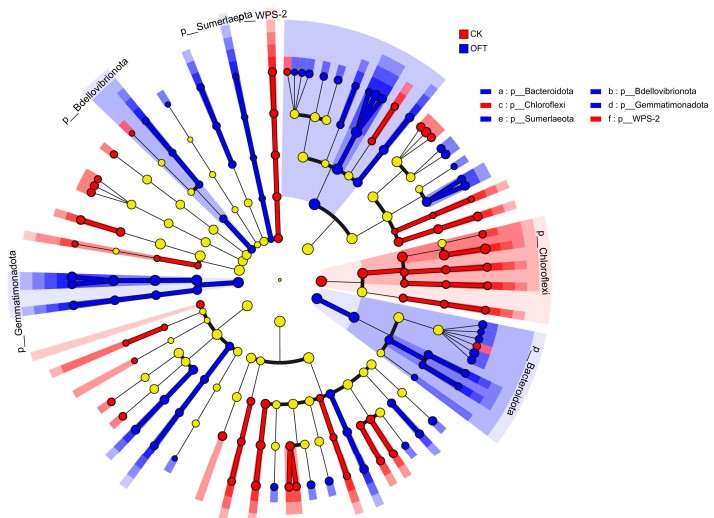

**Figure 5.** LEfSe analysis at multiple bacterial taxonomic levels indicating the difference among optimized fertilization treatment (OFT) and the control (CK). Red, blue, and green dots indicate taxa significantly enriched in CK and OFT treatments, respectively. Yellow dots represent the bacteria taxa without difference among treatments.

For soil fungi, according to LEfSe analysis (LDA threshold 2), it can be seen (Figure 6) that there are 18 and 12 taxa contributing significantly to the differences in CK and OFT treatments, respectively. Mortierellomycota represented biomarker in the control soil at phylum level (Figure 6A). The genera *Acremonium*, *kernia*, *Mortierella*, *Fusarium*, and *Phialosimplex* contributed significantly to the differences between the treatment and control ($p < 0.05$) (Figure 6B).

### 3.3.3. Functional Prediction of Soil Bacterial Groups

PICRUSt analysis was used to predict the potential function of soil bacteria under different treatments (Figure 7). The figure lists the major metabolic pathways, and the genes related to amino acid metabolism, carbohydrate metabolism, energy metabolism, and xenobiotics biodegradation and metabolism in soil samples treated with optimized fertilization were all higher than those in the control group (Figure 7A). Further analysis of energy metabolic pathways showed that predicted metabolic pathways associated with oxidative phosphorylation, carbon fixation, methane metabolism, fatty acid metabolism, and nitrogen and sulfur metabolism were higher in optimized fertilization treatment, whereas metabolic pathways associated with photosynthetic processes were higher in control soils (Figure 7B). In general, balanced fertilization is beneficial to improve the metabolic process of soil bacteria.

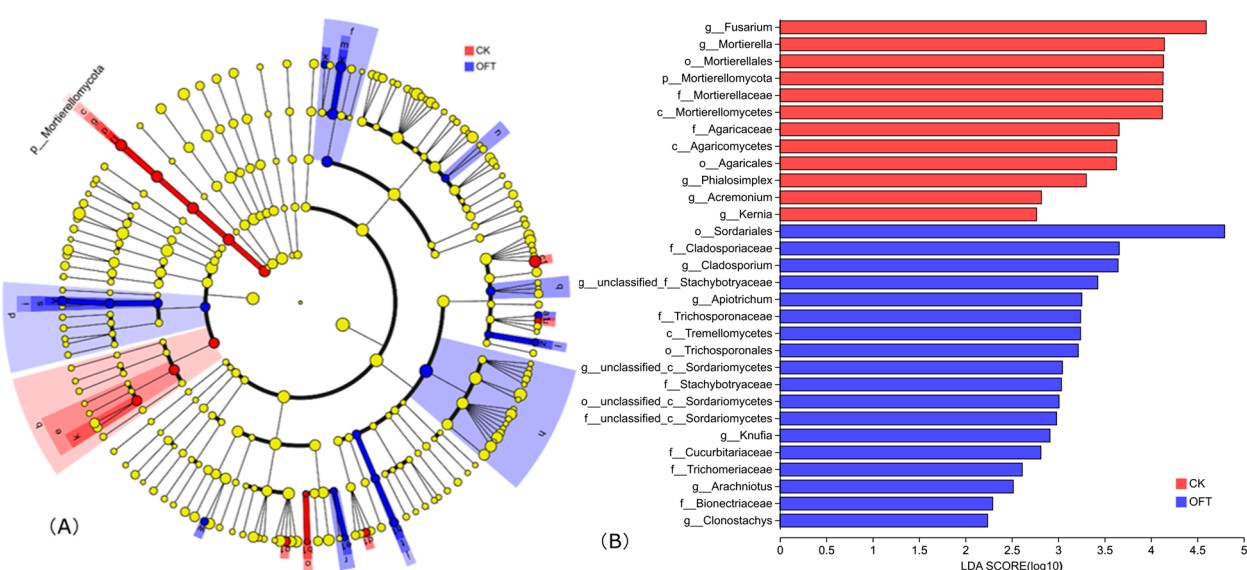

**Figure 6.** (**A**,**B**) LEfSe analysis at multiple bacterial taxonomic levels indicating the difference among optimized fertilization treatment and control. Red, blue, and green dots indicate taxa significantly enriched in CK and OFT treatments, respectively. Yellow dots represent the bacteria taxa without difference among treatments.

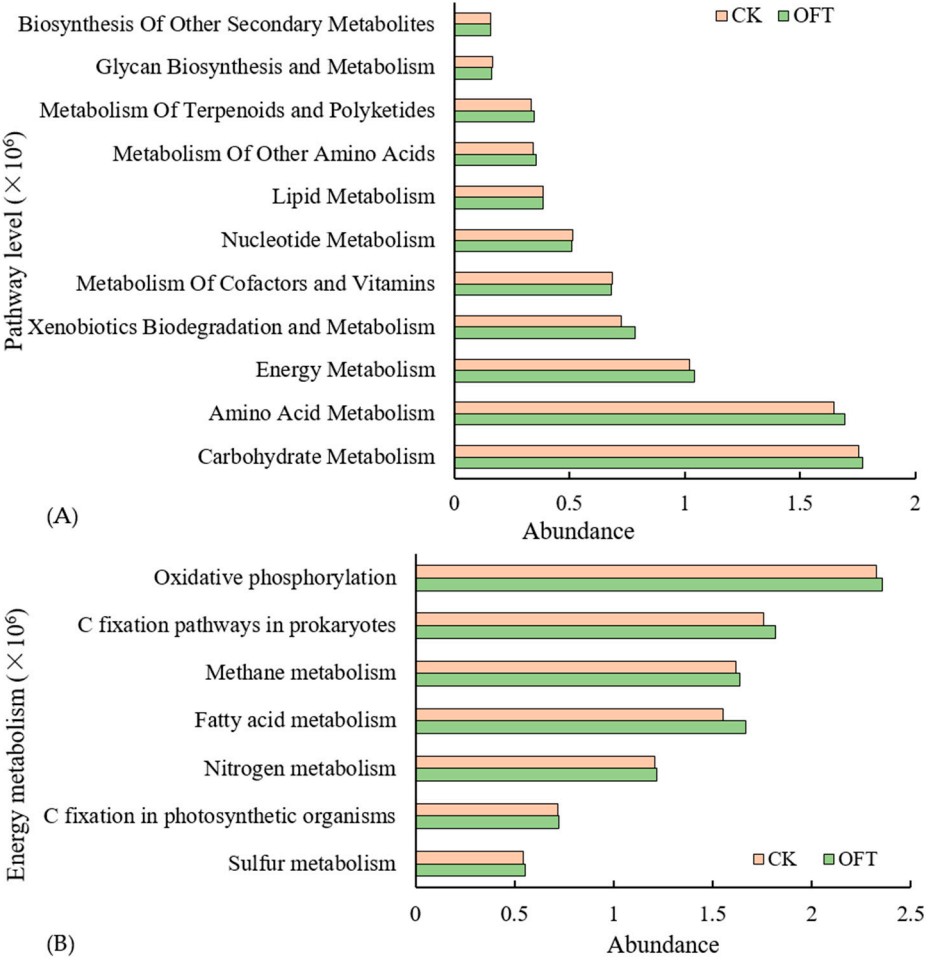

**Figure 7.** Potential functional gene abundance of bacteria participating in metabolic pathways (**A**) and energy metabolic pathways (**B**) in different treated soils, as predicted by KEEG.

### 3.4. Relationship between Soil Environmental Factors and Microbial Groups

Spearman correlation analysis was used to evaluate the relationship between the relative abundance of each microbial taxa in the samples and environmental factors (Table 3). There was a positive correlation between Gemmatimonadota and Armatimonadota and soil-available Mo content and pH, respectively. However, Deinococcota and Abditibacteriota were negatively correlated with soil available calcium, magnesium, boron contents, and EC. Most of the phyla significantly affected by soil factors were non-dominant taxa (relative abundance < 1%).

**Table 3.** Person correlations between the relative abundance of major bacterial and fungi phyla and soil physiochemical properties.

|  | Phylum | pH | EC | NN | Ca | Mg | B | Mo |
|---|---|---|---|---|---|---|---|---|
| Bacteria | Patescibacteria | −0.49 | −0.77 | −0.66 | −0.89 * | −0.77 | −0.77 | −0.54 |
|  | Gemmatimonadota | 0.09 | 0.37 | 0.43 | 0.54 | 0.37 | 0.60 | 0.94 ** |
|  | Verrucomicrobiota | 0.31 | −0.77 | −0.89 * | −0.60 | −0.77 | −0.71 | 0.03 |
|  | unclassified_k__norank_d__Bacteria | −0.03 | −0.75 | −0.84 * | −0.81 * | −0.75 | −0.90 ** | −0.75 |
|  | Abditibacteriota | −0.47 | −0.88 * | −0.79 | −0.97 ** | −0.88 * | −0.88 * | −0.50 |
|  | Armatimonadota | 0.89 * | 0.31 | 0.09 | 0.37 | 0.31 | 0.14 | 0.14 |
|  | Deinococcota | −0.31 | −1.00 *** | −0.94 ** | −0.94 ** | −1.00 *** | −0.89 * | −0.20 |
| Fungi | Chytridiomycota | −0.09 | 0.60 | 0.77 | 0.66 | 0.60 | 0.83 * | 0.43 |
|  | unclassified_k__Fungi | −0.14 | 0.83 * | 0.94 * | 0.71 | 0.83 | 0.83 | 0.14 |
|  | Rozellomycota | 0.49 | 0.77 | 0.66 | 0.89 * | 0.77 | 0.77 | 0.54 |

Only bacterial and fungal phyla with significant correlation with soil parameters were listed. EC: electrical conductivity; NN: nitrate-N; Ca: water-soluble calcium; Mg: water-soluble magnesium; B: available boron; Mo: available molybdenum. * $0.01 < p \leq 0.05$, ** $0.001 < p \leq 0.01$, and *** $p \leq 0.001$.

### 3.5. Effects of Different Treatments on the Growth Amount of Chinese Cabbage

The effects of optimized treatments on the growth of Chinese cabbage are shown in Figure 8. It can be seen from the figure that both fresh and dry biomass were increased after the balanced fertilization. The fresh weight of cabbage increased significantly by 27.1% ($p < 0.05$) by optimized fertilization, from 18.44 g per pot to 23.8 per pot.

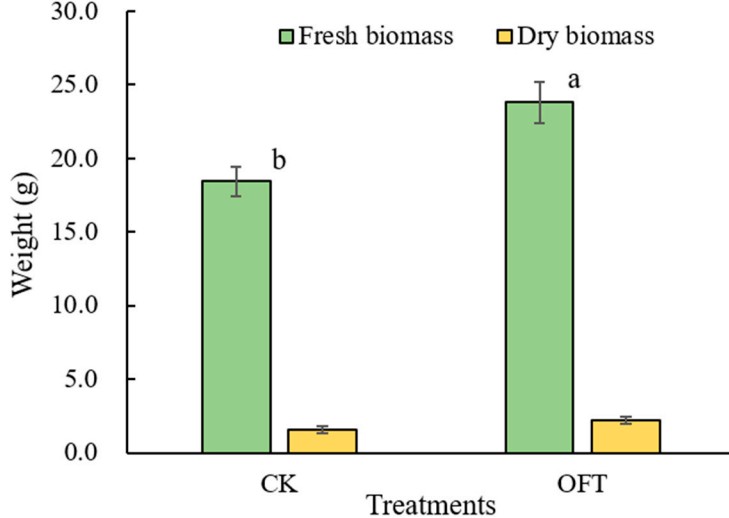

**Figure 8.** Fresh and dry weight of Chinese cabbage under different treatments. Different letters indicate significant differences among the treatments in biomass of Chinese cabbage.

## 4. Discussion

Nutrient imbalance and acidification are the most important soil obstacles in facility agriculture [12,37]. Despite the urgent problems caused by inappropriate fertilizer management, few studies have concerned regulating soil nutrient balance [38]. Here, an optimized fertilizer scheme was carried out, on the bases of soil physiochemical conditions. Special

attention was paid to the adjustment of soil acidity and the balance of soil's major, middle, and micro nutrients. The results showed that soil pH was increased with the application of calcium oxide and magnesium oxide, though it was not yet close to the optimal acidity range for plant growth. This indicated that the application dose of calcium oxide and magnesium oxide could still be increased. After optimized fertilization, the number of culturable bacteria, as well as bacterial diversity, increased, probably due to better growth condition [39].

After optimized fertilization, the unique bacterial taxa in soil increased, further proving that soil bacterial diversity increased. Optimized fertilization had a certain effect on the relative abundance of some dominant bacteria in soil. Proteobacteria, Firmicutes, Gemmatimonadota, and Bacteroidetes increased, whereas Actinobacteria, chloroflxi, and superphyla decreased. Among them, Actinobacteria are generally considered as soil beneficial bacteria because they can promote plant growth, improve crop stress resistance, and participate in soil CNPS nutrient cycling [40]. However, in the experiment, the relative abundance of Actinobacteria decreased slightly after optimized fertilization but did not reach a significant level. As this group of microorganisms could thrive in harsh conditions [41], they might not be sensitive to the change in micronutrients. A large number of microorganisms are defined as plant promoting rhizobacteria, such as *Bacillus*, *Arthrobacter* [42], *Sphingomonas* [43], and *Gemmatimonas* [44], that are beneficial for plant growth through regulating soil carbon, nitrogen, phosphorus cycles, or improving plant resistance. In our experiment, the optimized fertilization significantly enriched these microorganisms, which might have contributed to the increase in cabbage growth.

The analyses of soil fungal community revealed that the optimized fertilization resulted in increased fungal diversity, whereas the abundance of culturable fungi decreased significantly. Previous studies showed that continuous facility cultivation would lead to a decrease in bacterial and fungal diversity [28,29], due to various soil obstacles as a result of improper fertilization. After regulating soil conditions with optimized fertilization, this problem could be solved to some extent. The abundance of some common pathogenic groups in the treatment soil was significantly higher than that in control, such as *Fusarium*, one of the most important pathogens in ascomycetes, which often causes plant wilt in the soil of obstacle facilities [45]. The decline of this type of bacteria indicates that optimized fertilization can effectively reduce the damage of soil pathogens. *Sordariales*, within the phylum Ascomycota, known as beneficial fungi in soil that could have positive effects on the formation of soil aggregates [42], was also enriched by optimized fertilization. Other important biomarkers, including *Mortierella,* were found to be significantly higher in the control soil than the optimized fertilizer treatment. Known as one of the most abundant fungal groups in agricultural soils, this genus is reported to take part in improving P and Fe availability, as well as the release of organic C [46]. They are also found to thrive in acidic soils [47], which might explain the relatively lower abundance in the optimized fertilizer treatment.

Changes in microbial community structure may further affect soil fertility and crop growth by regulating soil pH and other nutrient cycling processes [48]. It was found that the abundance of various metabolic pathways, as well as the energy metabolism in the soil, increased under the condition of balanced fertilization, which proved that the microbial function was developing in a favorable direction [49]. This effect is probably related to crop growth, and the soil nutrient cycling process mainly depends on microbial activities, which ultimately ensures the high yield of Chinese cabbage.

Correlation analysis showed that only the less abundant bacterial and fungal phyla were correlated to soil properties, indicating that the rare communities were more sensitive to environmental changes [50]. Rare taxa are often neglected and considered ecologically non-relevant, but they might become important after the change in environmental conditions [51]. In this study, the rare species respond quickly to the treatment of optimized fertilization. It can be predicted that continuous long-term balanced fertilization can cause

dramatic change in soil microbial community and thus help to recover the soil ecosystem in facility agriculture, where soil obstacles are common.

The improvement of soil nutritional conditions and microbial community composition as a result of optimized fertilization eventually increased vegetable production. It is well known that long-term facility production has caused serious soil problems, including not only an imbalance of nutrients but also an accumulation of toxic substances. Therefore, it is also important to control the safety of vegetable production and to study the quality of vegetables under optimized fertilization in the future.

## 5. Conclusions

This study explored the effects of optimized fertilization in facility agriculture. The results demonstrated that soil acidification was relieved by the addition of calcium and magnesium oxides. Balanced fertilization decreased soil EC and increased the critical micro-nutrients B and Mo. Bacterial diversities were increased, with more beneficial microorganisms enriched, such as *Bacillus*, *Arthrobacter*, and *Gemmatimonas*, whereas the abundance of fungi decreased, and the common panthogenic species, including *Fusarium*, were significantly reduced after optimized fertilization. The predicted bacterial metabolic pathways were also increased. The biomass of Chinese cabbage was also significantly promoted, by 27.1%. Therefore, accurate fertilization based on soil analyses is recommended for the sustainable development of facility agriculture.

**Author Contributions:** Conceptualization, Y.L., X.Q. and J.W.; methodology, Y.L.; software, G.W.; validation, Y.L., J.W. and X.Q.; formal analysis, Y.L., Y.M. and J.W.; investigation, Y.L. and Y.M.; resources, X.Q. and J.W.; data curation, G.W. and J.W.; writing—original draft preparation, Y.L. and J.W.; writing—review and editing, X.Q.; supervision, J.W. and X.Q.; project administration, J.W.; funding acquisition, J.W. All authors have read and agreed to the published version of the manuscript.

**Funding:** This research was funded by the Key research and development projects (social development) of Yangzhou, grant number YZ2022060, Green, high quality and high efficiency vegetable production project (SNBN-2013-17), and the National Steering Committee of Agricultural Graduate Education project, grant number 2021-NYYB-13.

**Data Availability Statement:** The data presented in this study are available upon request from the corresponding author. The data are not publicly available due to compliance with data protection regulations.

**Conflicts of Interest:** The authors declare no conflict of interest.

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
