# Peer review of "Optimized Fertilization Shifted Soil Microbial Properties and Improved Vegetable Growth in Facility Soils with Obstacles"

_horticulturae, doi:10.3390/horticulturae9121303_

Round 1

Reviewer 1 Report

Comments and Suggestions for Authors

MDPI

Horticulturae Editorial

I am sharing my comments about the manuscript “horticulturae-2726682: Optimized Fertilization Shifted Soil Microbial Properties and Improved Vegetable Growth in Facility Soils with Obstacles”. The manuscript showed the effects of fertilization on soil attributes. There was an emphasis on microorganisms in the soil. In the material and methods, there was a lack of information about soil texture, calcium and magnesium content in soil, and experimental design. The treatments aren’t clear in the description of the experimental setup. Furthermore, the authors didn’t write if the experiment was in blocks or completely randomized. In my opinion, some results could be better if a “means test” was done. Doubts and suggestions are in the manuscript.

Best regards!

Author Response

Answers to the reviewer’s comments

Reviewer 1

Q1. Soil texture?

A1. The soil is sandy loam

Q2. Did you have Calcium and Magnesium content?

A2. Yes, we have determined water soluble Ca and Mg and included in the method section.

Q3. What were the treatments? What was experimental design? Could you write about?

A3. The optimized treatment (OFT) is described in the second paragraph of section 2.2. We have included (OFT) now.

Q4. Why didn't you do a "Means Test"?

A4. We have done a Means test now and also included in the figures.

Q5. Can you write this without a "Test Means

A4. Please refer to Q4. We have performed Measns test and included the test result.

Q6. Where is the meaning of these abbreviation (CK and OFT)?

A6. Please refer to Q3. We have added the abbreviation to the method section.

Q7. Why didn't you do a "Means test"? If you had done a "Means test," you could write the letters in the treatments, and you would have a certain answer from your results (Figure 1)

A7. We have already included the test result in the figure.

Q8. 3.3.1 The highlighted part (might not be accurately presented).

A8. We have modified the highlight. ‘Community abundance’ to ‘bacterial community richness’; ‘community diversity’ to ‘bacterial community eveness’. The indices Table 2 were added.

Q9. Could you write a legend about the colors and the treatments? (Figure 2)

A9. We have modified the figures and included the explanation of colors in the legends.

Q10. The community bar composition.

A10. We have deleted the ‘bar’

Q11. Highlighted ‘prbole’ in Line 374 should be ‘problem’, corrected. 

‘longtime’ in Line 403 should be ‘long term’, corrected.

Reviewer 2 Report

Comments and Suggestions for Authors

Dear Authors, Attached is the article with corrections. Best regards

Comments on the Quality of English Language

Dear Authors, Attached is the article with corrections. Best regards

Author Response

Manuscript ID: horticulturae-2726682

Title: Optimized Fertilization Shifted Soil Microbial Properties and Improved Vegetable Growth in Facility Soils with Obstacles

Dear reviewer:

Thank you very much for your constructive comments and suggestions on our manuscript. We appreciate your effort and have revised the manuscript according to your suggestions.

Please see the attached revised manuscript with tracked changes (please be aware there are also revisions as suggested by other reviewers).

We would like to thank you again for the revision work!

Sincerely yours,

Juanjuan Wang

November 22, 2023 

Reviewer 3 Report

Comments and Suggestions for Authors

The paper presents the results of a pot experiment in which the impact of optimized fertilization on soil properties was assessed. The impact of optimized fertilization on selected soil chemical and microbiological properties was assessed. To assess microbial properties, a set of correctly selected analyzes such as the determination of culturable microorganisms in soil and soil microbial community structure.

The results presented in the study indicate that: balanced fertilization improved the assessed chemical properties of the soil, the diversity of bacteria increased, with more favorable microorganisms enriched organisms (Ballilus, Arthrobacter and Gemmatimonas), reduction in abundance mushrooms. an increase in predicted metabolic pathways was also demonstrated. Changes in the chemical and microbiological properties of the soil resulted in a significant increase in the biomass of Chinese cabbage. The conclusions presented in the work include recommendations for facility agriculture. They indicate that for sustainable development, fertilization determined on the basis of soil analyzes should be used.

Notes for authors:

·       provide the source of the data in lines 102-105,

·       check whether the data "A plastic box (305×200×90mm) was used for the experiment, with 5 kg of soil in each box (line 116) was entered correctly,

·       proposes changing the title of subsection 3.1 to: Effects of different treatments on the basic chemical properties of soil

·       provide how the bacterial diversity indexes were determined.

Author Response

Answers to the reviewer’s comments

Reviewer 2

Q1. provide the source of the data in lines 102-105

A1. These data were determined by ourselves, as reference for fertilization optimization. We have modified the corresponding sentences in method section. ‘The basic properties of soil were determined before the pot experiment:…’

Q2.  check whether the data "A plastic box (305×200×90mm) was used for the experiment, with 5 kg of soil in each box (line 116) was entered correctly

A2. We have checked and revised the sentence to ‘Plastic boxes (305×200×90mm) were used for the cultivation in the experiment, with 5 kg of soil placed in each box.’

Q3.  proposes changing the title of subsection 3.1 to: Effects of different treatments on the basic chemical properties of soil

A3. Changed as suggested.

Q4.  provide how the bacterial diversity indexes were determined.

A4. We have revised the sentence in the second paragraph of section 2.3.3. ‘Alpha (α) diversity of bacteria and fungi, including Sobs, ACE, Chao, Shannon and Simpson indices were calculated using formulas on Major I-Sanger platform.’